# Probabilities of developing HIV-1 bNAb sequence features in uninfected and chronically infected individuals

Christoph Kreer [1,15], Cosimo Lupo [2,14,15], Meryem S. Ercanoglu[1,15], Lutz Gieselmann[1,3], Natanael Spisak[2], Jan Grossbach [4], Maike Schlotz[1], Philipp Schommers [1,3,5,6], Henning Gruell [1,5], Leona Dold[7,8], Andreas Beyer [4,6], Armita Nourmohammad [9,10,11,12,13], Thierry Mora[2,16], Aleksandra M. Walczak[2,16] & Florian Klein [1,3,6,16] ✉

HIV-1 broadly neutralizing antibodies (bNAbs) are able to suppress viremia and prevent infection. Their induction by vaccination is therefore a major goal. However, in contrast to antibodies that neutralize other pathogens, HIV-1-specific bNAbs frequently carry uncommon molecular characteristics that might prevent their induction. Here, we perform unbiased sequence analyses of B cell receptor repertoires from 57 uninfected and 46 chronically HIV-1- or HCV-infected individuals and learn probabilistic models to predict the likelihood of bNAb development. We formally show that lower probabilities for bNAbs are predictive of higher HIV-1 neutralization activity. Moreover, ranking bNAbs by their probabilities allows to identify highly potent antibodies with superior generation probabilities as preferential targets for vaccination approaches. Importantly, we find equal bNAb probabilities across infected and uninfected individuals. This implies that chronic infection is not a prerequisite for the generation of bNAbs, fostering the hope that HIV-1 vaccines can induce bNAb development in uninfected people.

The adaptive immune system is able to cope with a plethora of different antigenic structures by employing a diverse repertoire of lymphocytes, which express unique immune receptors as a consequence of V(D)J recombination during lymphopoiesis[1]. B cell receptors (BCRs) further diversify during affinity maturation by somatic hypermutation (SHM)[2,3], leading to the generation of antibodies with high affinities that can target and neutralize infectious pathogens.

The human immunodeficiency virus-1 (HIV-1), however, is able to outpace the adaptive immune system by quickly evolving into antigenically diverse quasispecies due to its error-prone replication machinery[4]. These quasispecies contain viral variants that can escape from autologous circulating antibodies. As a consequence, the immune system adapts to the emerged variants, resembling an ongoing immunological arms race[5]. Notably, there is a rare fraction of HIV-1-infected individuals who develop a broad serum neutralization response against numerous viral variants, and from whom monoclonal broadly neutralizing antibodies (bNAbs) have been isolated[6-9] (reviewed elsewhere)[10-13]. These antibodies target various sites on the homotrimeric envelope glycoprotein (Env), including the CD4 binding site (CD4bs), the variable loop 1 and 2 apex region (V1/V2-apex), the V3 loop base with its surrounding glycans (V3 loop), the gp120/gp41 interface region with the fusion peptide, the membrane-proximal external region (MPER) of gp41, and the so-called 'silent-face'[14]. Importantly, bNAbs are able to prevent and treat HIV-1 as well as chimeric simian-(S)HIV-1 infections in animal models[15-20], and have been demonstrated to suppress viremia in HIV-1-infected individuals without notable adverse events or side effects[21-30]. For instance, a combination of the CD4bs antibody 3BNC117 and the V3 loop antibody 10–1074 was able to control viremia in 76% of study participants for at least 20 weeks in the absence of antiretroviral therapy (ART)[27].

Moreover, preventive treatment with the CD4bs bNAb VRC01 significantly reduced infections with VRC01-sensitive HIV-1 strains[31], demonstrating that bNAbs are in principle able to prevent infections in humans. Importantly, technical advances and efforts to screen thousands of HIV-1-infected individuals have promoted the isolation of nearly pan-neutralizing bNAbs[9,32,33]. This next generation of bNAbs, engineered bi- or trispecific antibodies[34–37], and combination therapies with complementary bNAbs[25,26,29] have the potential to constrict viral escape and thus improve HIV-1 treatment and prevention strategies.

Despite the progress in passive administration, all efforts to induce highly potent bNAbs through vaccination have failed so far. bNAbs tend to accumulate unusual sequence properties, including high numbers of somatic mutations[6–9,32,33,38], insertions/deletions[6,8,9,38,39], distinct $V_H$ gene segment use[40–42], or exceptionally long complementarity determining regions (CDRs)[7]. Long CDRH3 regions as well as the usage of $V_H 4$–$34$ have previously been associated with self-reactive antibodies in autoimmune diseases[43–45] and several bNAbs proved indeed to be auto- and polyreactive[46,47]. Since auto-reactive B cells are counter-selected during B cell development[45], it has been speculated that bNAb development is normally blocked by immune checkpoints that can only be bypassed through chronic infection[48]. In line with this, BCRs of chronically Hepatitis C virus (HCV)-infected individuals show increased CDRH3s lengths (i.e. potentially higher self-reactivity) in comparison to uninfected controls[49]. In addition, many potent HIV-1 bNAbs have been isolated after several years of infection[9,42,50], suggesting that a prolonged virus-antibody co-evolution is a requirement for their induction. Guided vaccine design attempts to mimic this co-evolution by serial immunizations with varying immunogens[51,52]. Yet it is currently unclear which bNAbs have the highest chance to be elicited and should thus be selected for these strategies. Previously, precursor frequencies of VRC01-like bNAbs and BG18 have been estimated in uninfected individuals by CDRH3 similarity searches[52–54] and probabilities of distinct mutations have been determined for a subset of bNAbs[55]. However, comprehensive methods and analyses that compare the combined probabilities of V(D)J recombination and overall mutation patterns across different bNAb classes are still lacking and it remains elusive how these probabilities are influenced by chronic infection.

Here, we performed unbiased next-generation sequencing (NGS) and learned probabilistic models for somatic point mutations and V(D)J recombinations on BCR repertoire data from 57 uninfected individuals. We applied these models to determine and compare the generation probabilities of 70 bNAbs. By correlating probabilities with neutralization efficacies, we identified broad and potent candidate bNAbs that are more likely to be elicited than others and are thus particularly suited as targets for vaccination strategies. Finally, we sequenced 34 HIV-1- and 12 hepatitis C virus (HCV)-infected individuals, to infer repertoire characteristics and learn models to determine bNAb generation probabilities in the presence of chronic infections.

## Results

### Performing unbiased sequencing of the B cell receptor repertoire

In order to determine and compare the probability of developing an antibody with a specific sequence, we aimed to establish a pipeline for collecting unbiased BCR repertoire data from peripheral B cells and infer antibody sequence statistics with high confidence (Fig. 1a). To this end, we set up a sorting strategy to isolate naive or IgG-class-switched (i.e. antigen-experienced) B cells from peripheral blood mononuclear cells (PBMCs, Supplementary Fig. 1) and developed a 5′-rapid amplification of cDNA ends (RACE)-based sequencing protocol including unique molecular identifiers (UMIs) for computational error correction (Fig. 1a).

To test whether this sequencing approach yields sufficient high-quality reads, we analyzed biological duplicates of 100,000 IgG$^+$ B cells from three blood donors (Fig. 1b). From 1,354,097 to 2,552,903 raw reads per replicate, we reconstituted on average 6121 IgG heavy, 18,868 kappa, and 12,000 lambda chains after filtering for error-corrected and productive sequences (Supplementary Fig. 2, Supplementary Data 1). There was substantial overlap in unique CDRH3s within samples from the same individuals (on average 2.4–4.1% for biological replicates and 81% for the technical replicate, Fig. 1c) in comparison to the overlap between different donors (<0.02%, Fig. 1c, lower panel). Moreover, repertoire features such as CDRH3 length distribution (Fig. 1d) or $V_H$ gene mutation frequencies (Fig. 1e) were indistinguishable between biological replicates, but significantly differed across individuals. To estimate the resolution of the sequencing approach, we spiked in varying concentrations (0.01–10%) of two B cell lymphoma (BCL) cell lines into naive B cells from a blood donor. Spiked-in cells could be detected at all concentrations, including as few as 10 in a total of 100,000 cells (Supplementary Fig. 3). Concluding that the sequencing pipeline yields reproducible and representative statistics, we collected samples from 57 healthy (i.e., not infected with HIV-1 or HCV) individuals that comprised 54% male and 46% female donors with a mean age of 33 years (Fig. 1f). By sorting and sequencing in total 5,700,000 IgG positive B cells we generated 96,825,014 raw reads, yielding high-quality productive sequences for 287,505 heavy, 839,776 kappa, and 476,835 lambda chains, with on average 5044 heavy, 14,733 kappa, and 8366 lambda chains per individual (Fig. 1g, Supplementary Data 2). Although sequence features were predictive of the repertoire's origin (Fig. 1d, e), they are relatively conserved between individuals on a global level (Fig. 1h). In accordance with previous studies[56–58], we find particular V gene segments (such as $V_H 1$–$69$ or $V_H 3$–$23$) to be highly abundant in our cohort, CDR3 lengths to average around 15–16, 9, and 11 amino acids and average mutational loads to peak around 7, 4, and 4% nucleotide V gene mutation frequency for heavy, kappa, and lambda chains, respectively.

We conclude that the presented pipeline allows for inferring high-quality repertoire statistics from a starting material of 100,000 B cells, and that this sample is representative of the unique IgG BCR repertoire feature distributions of a single individual. On average, IgG repertoires show conserved sequence feature distributions that reflect different probabilities for specific sequence characteristics to develop during B cell development and maturation.

### Predicting bNAb probabilities for V(D)J recombination and somatic hypermutation

To predict the probabilities of developing HIV-1 specific bNAbs, we retrieved sequence and neutralization data for 70 HIV-1 neutralizing antibodies with varying neutralization breadths and potencies against a panel of 56 different HIV-1 strains (Supplementary Fig. 4 and Supplementary Data 3) from the CATNAP database[59].

The 70 antibodies target various sites on the envelope spike-protein (Fig. 2a, adapted from Klein et al.[11], as well as Sok and Burton[60]) and cover a broad range of potencies with geometric mean IC$_{50}$ values from 0.005 to 16.991 μg/ml as well as neutralization breadths between 10.7 and 98.2% (Supplementary Data 4, Fig. 2b). Breadth and potency are the most critical parameters of HIV-1 neutralizing antibody activity and can diverge substantially among bNAbs. For example, the MPER bNAb 4E10 shows an exceptional breadth with a modest potency, while the CD4bs bNAb CAP256-VRC26.25 is highly potent but inferior in breadth (Fig. 2b). Moreover, bNAbs are often tested against differing viral strains, leading to deviating measures of breadth and potency, which complicates their comparability (Supplementary Fig. 4). To solve these problems and directly compare and rank bNAbs by their overall neutralization efficacy, we extracted a representative subset of 56 viral strains from the 118 panel reported by Seaman et al.[61] against which all 70 selected antibodies have been tested (Supplementary

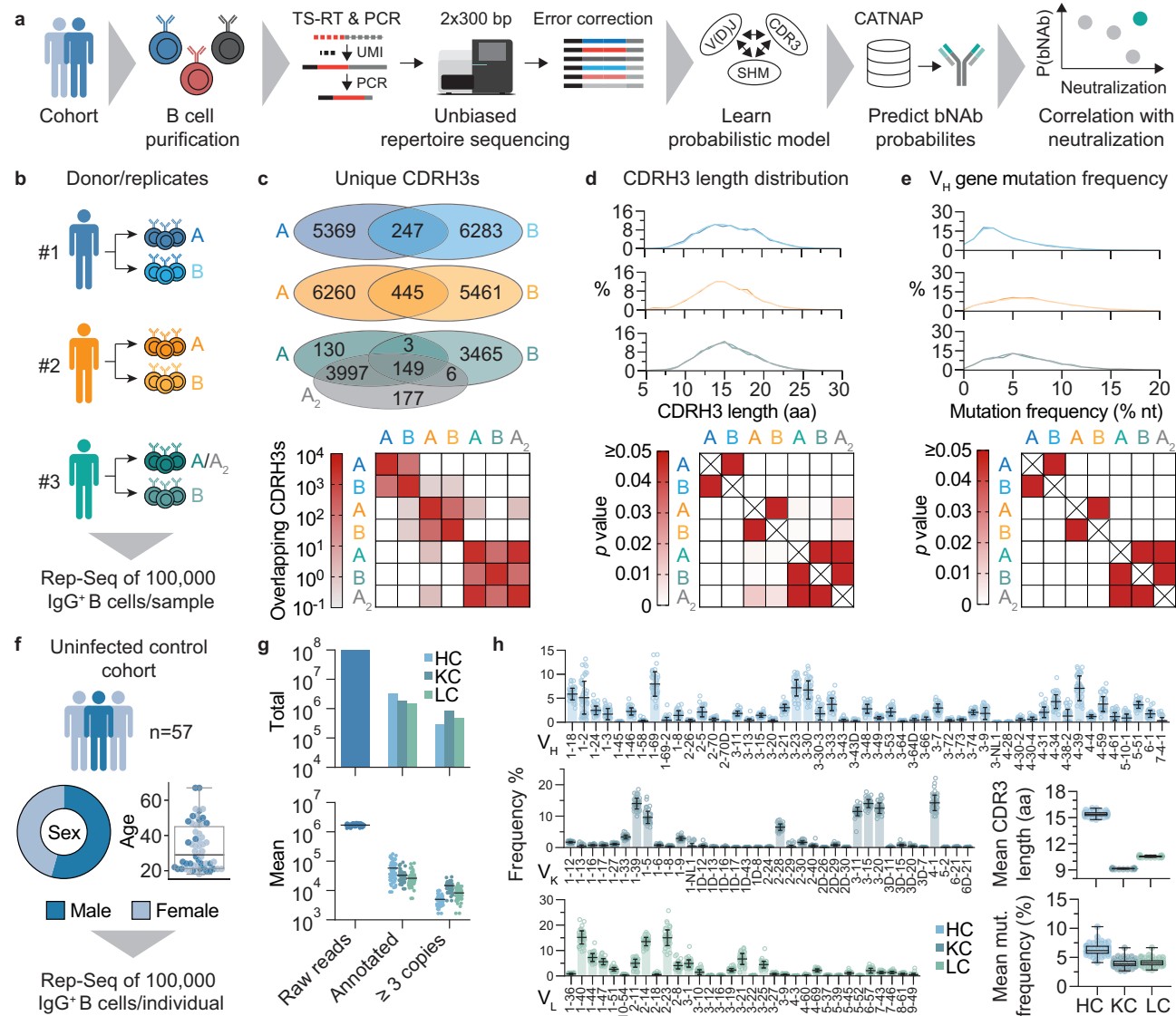

**Fig. 1 | Overall approach and unbiased repertoire sequencing. a** B cells are purified from peripheral blood and mRNA is isolated. Unique molecular identifiers (UMI) are added by template switching reverse transcription (TS-RT) and variable regions are amplified by PCR. Amplicons are sequenced by 2x300 bp Illumina sequencing and UMIs are exploited for error correction. High-quality sequences are used to learn probabilistic models and predict probabilities of bNAb sequences from the CATNAP database. Correlation of probabilities and neutralization allows for identifying highly probable and potent bNAbs. **b** Biological replicates of 100,000 IgG+ B cells were isolated from three uninfected donors for pipeline validation. $A_2$ represents a technical replicate of A (i.e., the same PCR product, but independent library preparation and sequencing). **c** Overlap of unique CDRH3s between replicates from samples in **b**. Upper panel shows the total number of overlapping CDRH3s between biological and technical replicates. Lower panel shows CDRH3 overlap within and between donors as the mean overlap of $n = 3623$ randomly drawn CDRH3s from each dataset after 100 iterations (see methods for details). **d** CDRH3 length distributions from samples in **b**. Upper panel shows

overlayed distributions for biological replicates. Lower panel shows $p$-values from Dunn post-hoc test after global Kruskal–Wallis test. **e** $V_H$ gene nucleotide (nt) mutation frequency distributions from samples in **b**. Upper and lower panels show overlayed distributions and $p$-values determined as in **d**. **f** Cohort sex and age distributions of $n = 57$ uninfected individuals for IgG+ repertoire sequencing. **g** Total and mean number of raw reads, annotated reads, and identical sequences (i.e., the same UMI) that were found at least three times in $n = 57$ uninfected individuals. **h** V gene segment distributions, mean CDR3 lengths, and mean V gene mutation frequencies for heavy, kappa, and lambda chains of $n = 57$ uninfected individuals. V gene segment distributions are depicted as mean values ± SD. Box-plots in **f** and **h** depict the 25% and 75% percentiles with the median as average lines and minimum/maximum values as whiskers. CDR complementarity determining region, SHM somatic hypermutation, Rep-Seq repertoire sequencing, HC heavy chain, KC kappa chain, LC lambda chain. Source data are provided as a Source Data file.

Fig. 4, Supplementary Data 3) and mathematically combined breadth and potency based on this panel into a single neutralization score (see method sections for more details; Supplementary Data 4, Fig. 2b). In terms of sequence characteristics, the 70 bNAbs show broad distributions for $V_H$ gene segment usage (Fig. 2c), CDRH3 lengths (Fig. 2d), and $V_H$ gene mutation frequencies (Fig. 2e), which differ substantially from the averaged IgG memory B cell repertoire statistics of the 57 uninfected individuals. Notably, separating bNAbs into

binding classes demonstrates that individual bNAbs are not necessarily extreme in all features. CD4 binding site antibodies, for example, often incorporate $V_H1-2$ or $V_H1-46$ and are highly mutated, while their CDRH3 lengths are within the range of the memory IgG reference distribution (Fig. 2c–e, blue bars). V2-apex antibodies, on the other hand, typically exhibit long CDRH3s, but are also less mutated (Fig. 2d, e, yellow bars). Both classes of antibodies can be found among the top neutralizing antibodies (Fig. 2b).

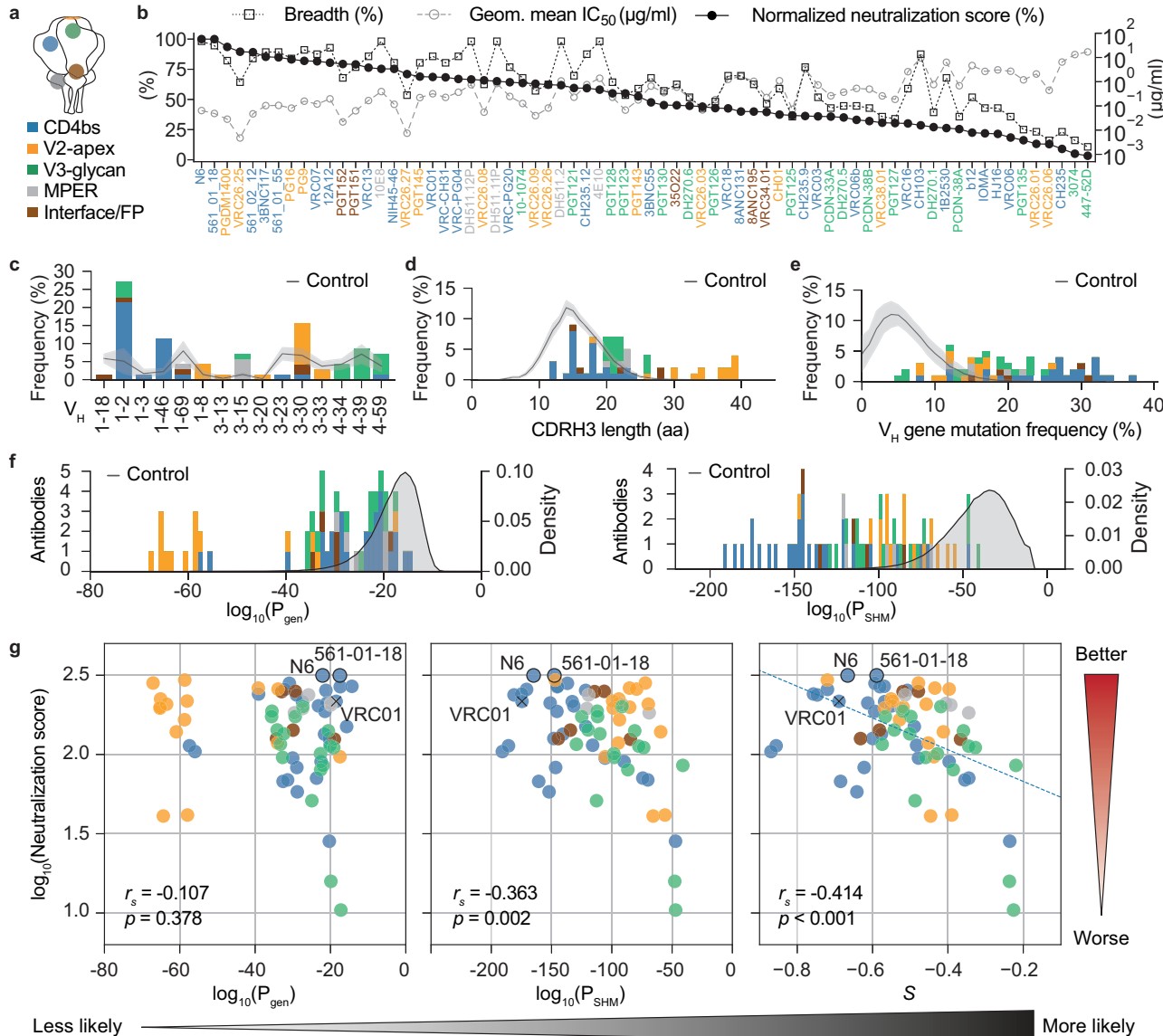

**Fig. 2 | Neutralization efficacy and sequence characteristics of broadly neutralizing antibodies targeting HIV-1. a** bNAb epitopes on the HIV-1 envelope spike. MPER membrane proximal external region, CD4bs CD4 binding site, FP fusion peptide. **b** Breadth, geometric mean half inhibitory concentration ($IC_{50}$) of neutralized strains, and normalized neutralization score for $n = 70$ bNAbs that have been tested against the same 56 HIV-1 strains. **c** Heavy chain V gene segment usage for the selected bNAbs. **d** Heavy chain CDR3 length distribution for the selected bNAbs in amino acids (aa). **e** Heavy chain V gene mutation frequency distribution for selected bNAbs. Controls in **c**–**e** depict the mean frequencies of the respective sequence features as solid lines ± SD as shaded areas from the $n = 57$ uninfected individuals (Fig. 1 f–h). **f** Heavy chain $P_{gen}$ and $P_{SHM}$ distribution for selected bNAbs derived through IGoR by a model learned on productive sequences from $n = 57$

uninfected individuals. Controls represent $P_{gen}$ and $P_{SHM}$ distributions for productive sequences from the $n = 57$ uninfected individuals. **g** Correlation plots of bNAb neutralization scores against heavy chain $P_{gen}$, $P_{SHM}$, and a combined probability score $S = c_1 \log_{10}(P_{gen}) + c_2 \log_{10}(P_{SHM})$, which was derived by a linear regression (dashed line) with $c_1 = 3.248 \times 10^{-03}$ and $c_2 = 3.604 \times 10^{-03}$. Spearman correlation coefficients $r$ and two-sided $p$ values are given in the figure. The correlation coefficient and two-sided $p$ value from the linear regression for $S$ are $r = -0.468$ and $p = 4.392 \times 10^{-05}$. Two near pan-neutralizing antibodies (N6 and 561-01-18) are highlighted by black outlines, one antibody that has been used for structure-guided vaccine approaches (VRC01) is highlighted by 'x'. Gradients on the right and bottom show directions for increasing neutralization activity and generation probability, respectively. Source data are provided as a Source Data file.

While long CDRH3s and high levels of somatic mutations could in part explain the rare occurrence of bNAbs, they do not account for heterogeneity in gene segment selection probabilities, or for any sequence context, which is known to bias V(D)J recombination and somatic hypermutation[62]. Moreover, simple descriptive statistics may suffer from undersampling of rare B cell sequences. We therefore applied the previously published Inference and Generation Of Repertoires (IGoR) tool[63], to provide quantitative and comprehensive estimates for the probabilities of generating a given CDRH3 ($P_{gen}$) and accumulating a unique pattern of point mutations ($P_{SHM}$). IGoR infers a probabilistic model that accounts for the statistics of V(D)J usage as

well as insertions and deletions in the junctional regions and sums over all generation scenarios consistent with the given BCR sequence to evaluate its overall generation probability. The SHM model learns the identities of mutated 5-mer subsequences. Importantly, estimating probabilities with models allows for overcoming limitations of sequencing depth through generalization. To investigate probabilities of bNAb sequence features after selection and affinity maturation, we took the combined quality-filtered and productive IgG sequences of all 57 uninfected individuals to learn the models and determined $P_{gen}$ and $P_{SHM}$ for the 70 bNAb heavy chain sequences (Fig. 2f, Supplementary Data 5). Almost all bNAbs show lower $P_{gen}$ and $P_{SHM}$ values (i.e., are less

likely) than the median uninfected IgG repertoire distribution, resembling the CDRH3 length and $V_H$ gene mutation frequency comparison in Fig. 2d, e. Indeed, there is a strong correlation between these sequence features and the probabilities (Supplementary Fig. 5), suggesting that CDRH3 length and numbers of mutations are strong determinants of antibody probabilities. However, there are also antibodies with equal CDRH3 lengths (e.g. VRC03/DH511.11P) or similar amounts of SHM (e.g. VRC06b/3BNC117) but substantially different probabilities (over several log steps), highlighting the contribution of additional factors such as biases in V(D)J recombination and SHM (Supplementary Fig. 5).

Given the broad distributions of $P_{gen}$ and $P_{SHM}$ for the different bNAbs, we asked whether they are correlated with neutralization efficacy, and whether we can identify highly potent bNAbs that are more likely to be generated. The neutralization score is not correlated with $P_{gen}$ but slightly correlated with $P_{SHM}$ alone for the 70 bNAbs (Fig. 2g left and middle panel; Spearman correlation $r_s = -0.107/p = 0.378$ and $r_s = -0.363/p = 0.002$, respectively). Performing a linear regression that takes into account the logarithms of $P_{gen}$ and $P_{SHM}$ yields a combined probability score $S$ (Fig. 2g right panel, Supplementary Data 5) that is highly predictive of the neutralization score ($r = -0.468$ and $p = 4.392 \times 10^{-5}$ for the linear regression, $r_s = -0.414/p < 0.001$ for Spearman correlation). Based on this regression, lower probability scores (i.e. less likely bNAbs) are correlated with better neutralization. Importantly, the correlation allows for identifying bNAbs that are highly potent, but easier to generate than others (Fig. 2g right panel, upper right corner). Similar results for the correlation of $P_{gen}$, $P_{SHM}$, and the probability score $S$ with the neutralization score were also obtained for light chains (Supplementary Fig. 6, Supplementary Data 5).

We conclude that the linear combination of the logarithms of $P_{gen}$ and $P_{SHM}$ (i.e. the combined probability score $S$) is highly predictive of the neutralization capacity with the overall tendency of less likely bNAbs to be the most potent ones. In addition, our modeling approach not only confirms that bNAbs are in general unlikely outcomes of the B cell development, but also provides a framework for ranking bNAbs by their overall sequence probabilities.

### Global BCR repertoire features under chronic infections are similar to uninfected repertoires with marginal differences

Since chronic infections have been described to interfere with B cell development and functionality[64], we wondered whether and to what extent they influence memory IgG BCR sequence characteristics. To answer this question, we sequenced BCRs from the IgG$^+$ memory compartment of 34 HIV-1 and 12 hepatitis C virus (HCV)-infected individuals and compared them to the 57 uninfected repertoires (Fig. 3a).

Processing and filtering of 54,628,577 HIV-1 and 19,767,705 HCV raw reads yielded in total 1,382,301 (HIV-1) as well as 486,488 (HCV) heavy and light chain sequences (Supplementary Data 6 and 7). Overall, sequence characteristics, including V gene segment usage, CDR3 length, and V gene mutation frequencies, were comparable to the uninfected control cohort for both chain types (Fig. 3b, c, d), although HCV-infected individuals showed slightly longer mean CDRH3 lengths (Fig. 3c). Similarly, we detected comparable average numbers as well as length distributions of heavy chain insertions and deletions (Supplementary Fig. 7a, b), except for marginally shorter insertions in HCV-infected versus HIV-1-infected individuals (Supplementary Fig. 7c). In terms of clonality, no obvious differences were observed in B cell clone tree structures (Fig. 3e) with respect to skewness (demonstrated by comparable distributions and distribution centers of the weights ratios $w_D/w_{anc.}$ and the branch lengths; Fig. 3f), size distribution (Fig. 3g), or diversity (Fig. 3h). Since antiretroviral therapy (ART) is suppressing virus replication and thus may dampen anti-HIV-1 immune responses, we stratified the HIV-1 cohort by

treatment (Supplementary Fig. 8). Whereas repertoires from treated HIV-1-infected individuals were indistinguishable from uninfected individuals, we found a substantially higher fraction of $V_H$ gene segments $V_H1$–$69$ and $V_H4$–$34$ in untreated HIV-1-infected individuals, which did not reach significance after correcting for multiple testing. In addition, mean CDRH3 lengths were slightly but significantly longer in untreated individuals. Mean V gene mutation frequencies did not differ significantly, although the untreated subgroup contains one individual with noticeably less somatic mutations, which is mainly responsible for the visible shift in the V gene mutation frequency distributions (Fig. 3d, Supplementary Fig. 8). We also measured serum-derived poly-IgG neutralization breadth against the 12-strain global panel[65] for 33 of the 34 HIV-1-infected individuals (Supplementary Data 8). Of note, broad serum neutralization is found in ART-naive (untreated) as well as ART-treated individuals of our cohort (Supplementary Data 8). Stratifying BCR repertoires by neutralization breadth into high (≥66%), intermediate (<66% and ≥33%), or low (<33% and >0%) breadth and no neutralization (0%) did not yield any evidence that serum neutralization breadth is linked to relevant global changes in BCR repertoire features (Supplementary Fig. 9).

We conclude that V gene segment usages and CDRH3 lengths are marginally skewed in untreated, chronically infected individuals. However, on average, repertoire features remain almost constant between the cohorts.

### Equal probabilities for bNAb development in uninfected and chronically infected individuals

To investigate whether chronic infection shifts the probability to develop distinct bNAb sequences, we used the IgG repertoire data from HIV-1 and HCV-infected individuals to train IGoR and learn models for $P_{gen}$ and $P_{SHM}$.

Whole repertoire $P_{gen}$ distributions of both cohorts were almost identical to the uninfected control data for heavy, kappa, and lambda chains (Fig. 4a). Repertoire $P_{SHM}$ distributions on the other hand were slightly shifted towards less negative values for HCV- and HIV-1-infected individuals (Fig. 4b), which resembles the decrease in $V_H$ gene mutation frequency that was already observed in the repertoire data (Fig. 3d). We speculated that marginal differences in these distributions could still influence the probability for certain rare V(D)J recombinations or mutational patterns of distinct bNAb sequences. Therefore, we calculated $P_{gen}$ and $P_{SHM}$ for the individual 70 bNAbs with the models that have been trained on the repertoire data from infected individuals (Supplementary Data 5). When comparing those to the values that have been inferred from the uninfected cohort, there is only little deviation in $P_{gen}$ or $P_{SHM}$ (Fig. 4c, d). Importantly, there is no tendency for a whole antibody epitope group to be favored in one or the other cohort. To finally compare the overall probabilities of individual bNAbs between the cohorts, we finally calculated the combined probability score (S) using the cohort-specific $P_{gen}$ and $P_{SHM}$ values together with the coefficients from the uninfected repertoire linear regression model of Fig. 2g (Supplementary Data 5). In line with the similar $P_{gen}$ and $P_{SHM}$ values (Fig. 4c, d), combined probability scores were almost identical between uninfected and infected individuals (Fig. 4e). Despite the slight differences in the repertoire characteristics of untreated individuals (Supplementary Fig. 8), treatment status had no influence on bNAb probability scores (Fig. 4f). Similarly, we found no global change in probability scores when stratifying by serum neutralization breadth (Fig. 4g). Taken together, these results suggest no substantial differences in the overall probability of bNAb generation in the absence or presence of chronic HIV-1 or HCV infection.

## Discussion

Broadly HIV-1 neutralizing antibodies can effectively prevent and treat HIV-1 infections in animal models[16,17,19,66] and are considered promising

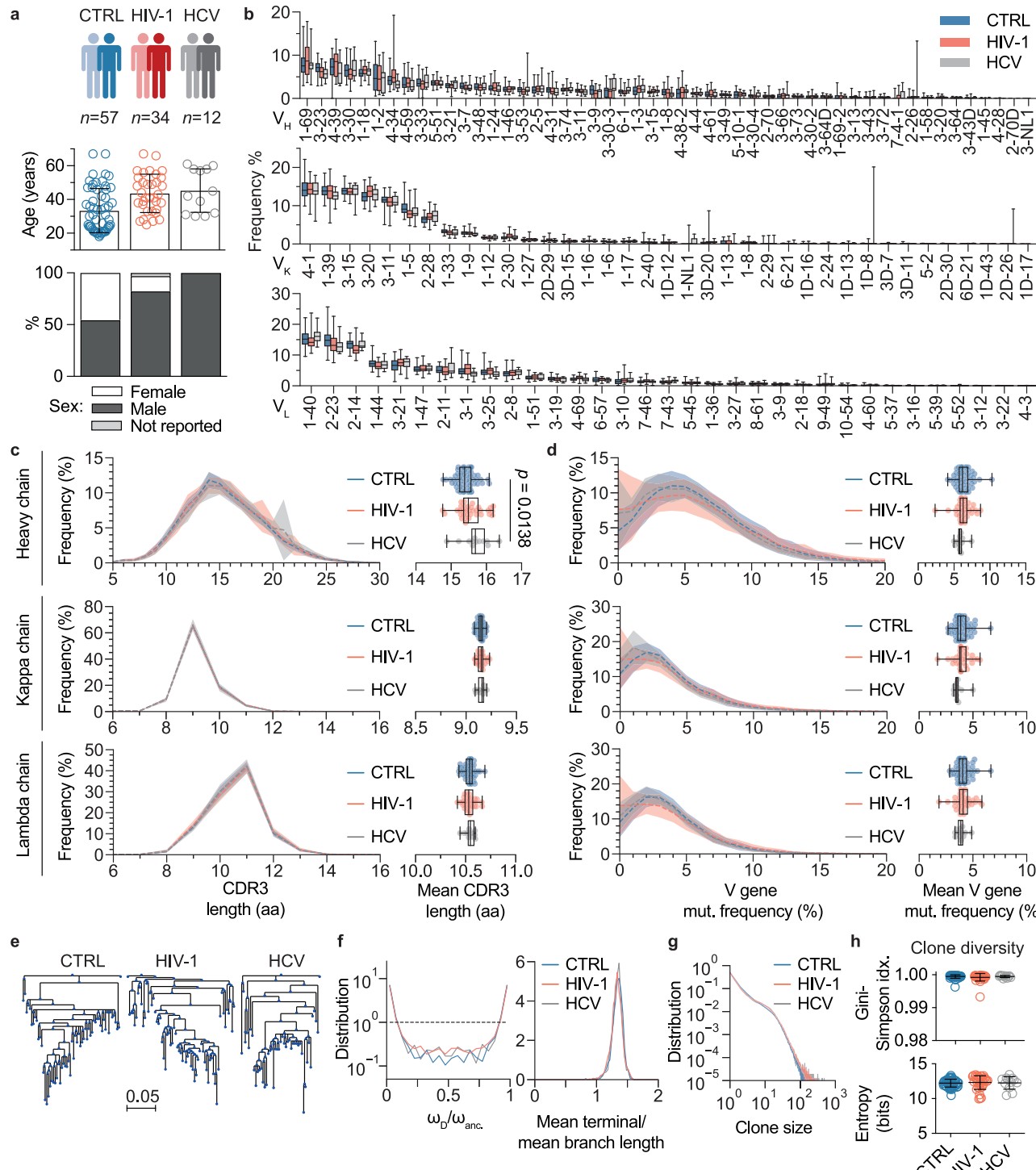

**Fig. 3 | IgG heavy chain repertoire characteristics of uninfected and chronically infected individuals. a** Cohort overview with age and sex distributions. Single dots in age distributions represent individuals, bar graphs and error bars depict mean cohort age ± SD. One individual (HIV-1) reported neither sex nor age. **b** V gene segment usage distributions for heavy, kappa, and lambda chains, ordered by descending frequencies according to the uninfected control (CTRL) group (*n* = 57 for CTRL, *n* = 34 for HIV-1, and *n* = 12 for HCV). **c** Mean CDR3 length distributions for heavy, kappa, and lambda chains in amino acids (aa). Left panel shows the mean CDR3 length distributions across individuals for each cohort (*n* = 57 for CTRL, *n* = 34 for HIV-1, and *n* = 12 for HCV) as solid lines and standard deviations as shaded areas. Right panel shows mean CDR3 amino acid length as dots for each individual and cohort statistics as box-plots. **d** Mean V gene nucleotide mutation (mut.) frequencies for heavy, kappa, and lambda chains. Representation of mean distributions (left panel) and individual means (right panel) as in **c** for all three cohorts

(*n* = 57 for CTRL, *n* = 34 for HIV-1, and *n* = 12 for HCV). One-way ANOVA with a two-sided Tukey-HSD post-hoc test was performed on the means in **c** and **d**. **e** Representative examples of clone trees for the cohorts based on heavy chain sequences. **f** The weights (i.e., the number of leaves deriving from a given node) were determined for the common ancestor of each clone ($\omega_{anc.}$) and its immediate descendants ($\omega_D$). Distributions of their ratios ($\omega_D/\omega_{anc.}$) were plotted (left panel) to illustrate the clone tree skewness (dashed line = neutral, see methods for details). The right panel shows the mean terminal/mean branch length distribution for clone trees. **g** Clone size distribution for the cohorts. **h** Clone diversity determined by Gini-Simpson index and entropy (*n* = 57 for CTRL, *n* = 34 for HIV-1, and *n* = 12 for HCV). Data is presented as mean values ± SD. Box-plots in panels **b**–**d** depict 25% and 75% percentiles with medians as average lines and minimum/maximum values as whiskers. Source data are provided as a Source Data file.

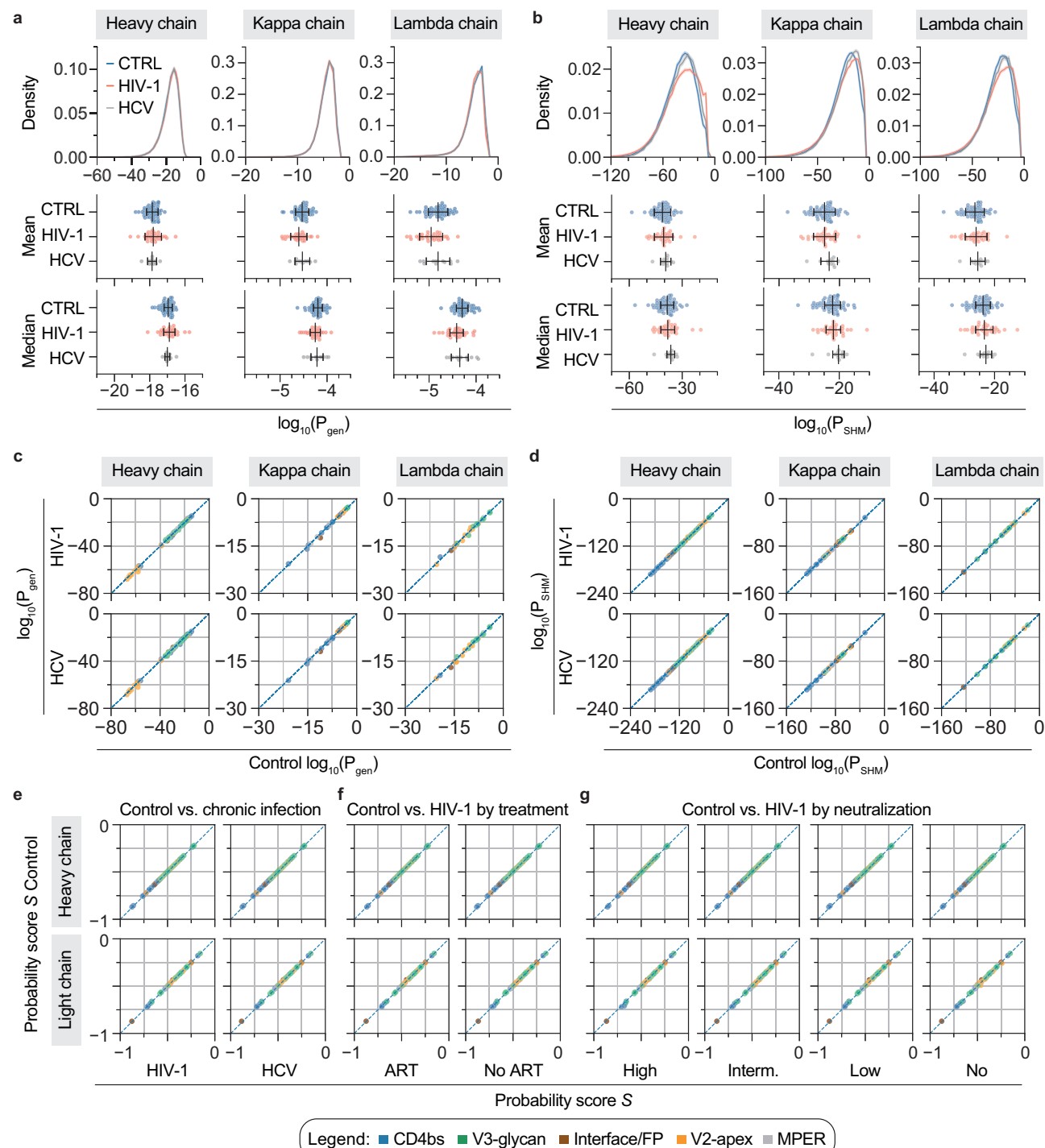

**Fig. 4 | Probability distributions and bNAb probability scores across cohorts.**
**a** $P_{gen}$ distribution as well as mean and median $P_{gen}$ for heavy and light chains derived from uninfected control (CTRL, $n = 57$ individuals), HIV-1-infected ($n = 34$ individuals), and HCV-infected ($n = 12$ individuals) cohorts by IGoR. Averages and error bars show the mean ± SD for mean $P_{gen}$ values and the median ± median absolute deviation (MAD) for median $P_{gen}$ values. **b** $P_{SHM}$ distributions for heavy and light chains for the same cohorts as in **a**. Averages and error bars show the mean ± SD for mean $P_{SHM}$ values and the median ± median absolute deviation (MAD) for median $P_{SHM}$ values. $P_{gen}$ and $P_{SHM}$ distributions in **a** and **b** show means as solid lines and SD as shaded areas. **c** Correlation plots of $P_{gen}$ for neutralizing antibody heavy ($n = 70$), kappa ($n = 33$), and lambda chains ($n = 26$) derived from either uninfected control (x-axis) or chronically infected cohorts (HIV-1, HCV, y-axis). **d** Correlation plots of $P_{SHM}$ for the same chains as in **c**. **e** Correlation plot for

bNAb heavy ($n = 70$) and light chain probability scores ($n = 59$) derived from HCV or HIV-1 cohorts (x-axis) in comparison to uninfected control (CTRL, y-axis). Cohort-specific $P_{gen}$ and $P_{SHM}$ of bNAbs were used to calculate the bNAb probability scores S using coefficients from the linear regression on uninfected individuals (Fig. 2g). The dashed line represents identity. **f** Comparison as in **e** with the HIV-1 cohort stratified by antiretroviral therapy (on ART, $n = 22$ individuals; off ART, $n = 12$ individuals). **g** Comparison as in **e** with the HIV-1 cohort stratified by serum neutralization breadth against the global HIV-1 panel into high (≥66%, $n = 8$ individuals), intermediate (≥33%, $n = 5$ individuals), or low (>0%, $n = 16$ individuals) breadth and no neutralization (0%, $n = 4$ individuals) to recalculate $P_{gen}$ and $P_{SHM}$ of bNAbs for these cohorts. Serum neutralization data for one individual was not available. ART anti-retroviral therapy. Source data are provided as a Source Data file.

tools for HIV-1 prevention and therapy in humans[67,68]. However, bNAbs have only been identified in a minor fraction of HIV-1-infected individuals and to elicit highly broad and potent bNAbs by vaccination remains a yet unreached goal[11,69,70]. The presence of high amounts of somatic mutations, insertions/deletions, or exceptionally long CDRH3s in bNAbs raised the questions of (i) whether these outstanding features explain their rareness, (ii) whether rareness is correlated with neutralization capacity, and (iii) if chronic infection is a prerequisite for their induction. By learning models from IgG[+] memory BCR sequences of 57 uninfected individuals, we estimated the probability for specific V(D)J recombinations ($P_{gen}$) and for accumulating patterns of somatic point mutations ($P_{SHM}$) in a representative set of HIV-1-neutralizing antibodies. We also tried to include insertions and deletions (indels). However, it was not possible to build a robust model, since their frequencies were too low in the repertoire data. In line with previous studies, we show that bNAbs accumulate improbable mutation patterns ($P_{SHM}$), which correlated with the binding site[55,71]. In particular, CD4bs antibodies had the least likely mutation patterns, while V2-apex and V3-glycan antibodies had the most likely ones. Similarly, we identified a subset of antibodies with highly improbable CDRH3 recombinations. These mainly comprise V2-apex antibodies that require long CDRH3s to penetrate the envelope glycan shield[72–74], but also some CD4bs antibodies with average CDRH3 lengths. These results support the hypothesis that bNAbs are rare, because the sum of their sequence features (i.e., either the CDRH3 sequence, the SHM patterns, or both) are unlikely to develop.

Previously, ongoing affinity maturation as well as the mean frequency of somatic mutations have been correlated with neutralization breadth and potency of bNAbs[75,76]. In line with this, we detected a weak correlation for the neutralization power of monoclonal bNAbs and SHM, looking not only at the frequency but also at the probability to accumulate a specific mutation pattern (i.e., $P_{SHM}$). Notably, we found that a linear combination of the logarithms of $P_{gen}$ and $P_{SHM}$ is predictive of bNAb neutralization efficacy, suggesting that bNAbs tend to be better, the more unlikely they are. Importantly, the correlation allows for identifying bNAbs that are highly potent but on average more likely than other, similarly potent ones. They include some of the above-mentioned V2-apex antibodies, which in contrast to their improbable CDRH3 showed more probable patterns of somatic hypermutations. Interestingly, V2-apex antibodies have been reported to occur relatively early during infection[77–79]. This suggests that the selection of rare precursors outperforms the accumulation of distinct mutations in timing and that V2-apex antibodies could thus be preferred over the heavily mutated CD4bs or the less potent V3 loop antibodies as targets for vaccination strategies. Of note, MPER antibodies also rank among the more likely bNAbs, although this class is less prevalent than e.g. CD4bs antibodies in the sera of HIV-1 infected individuals (up to 2/3, depending on the study)[80–84]. The lower prevalence has been previously attributed to a negative immunoregulatory control[85] due to the often poly- and autoreactive nature of MPER antibodies[46,47]. In our analysis, we determine antibody sequence features and do not account in general for antibody functions such as autoreactivity. However, antibodies or antibody sequences that are negatively selected would not be present in the analyzed memory B cell compartment (which was used to train the models) and therefore sequence features commonly associated with autoreactivity should be captured by this approach. While autoreactive properties of MPER antibodies are likely to limit their overall frequency, we conclude that on a sequence level they will be more probable in comparison to other sequence features, such as long CDRH3 or high levels of SHM found in V2-apex or CD4bs antibodies, respectively. The generation of MPER antibodies should therefore be less restricted and the higher probabilities of MPER bNAbs in the memory B cell compartment seem to be reasonable.

Notably, MPER antibody precursors have recently been successfully induced in the human HVTN133 immunization trial (NCT03934541)[86].

In addition to uninfected individuals, we also sequenced the repertoires of 34 HIV-1- and 12 HCV-infected individuals. In concordance with previous studies[49,87], we detected marginally longer mean CDRH3 lengths in HCV-infected as well as HIV-1-infected individuals that do not receive antiretroviral therapy. Similar to the findings from Roskin et al.[87], we also detected an increase of the inherently auto-reactive gene segment $V_H4–34$ in our ART-naive HIV-1-infected subgroup. However, we do not see substantial differences in bNAb sequence probability scores in the IgG[+] memory compartment when averaging over individuals in the cohorts. This suggests that chronic infection itself does not alter the probabilities to develop specific bNAbs substantially in any direction, even if it leaves traces of particular sequence features within the whole IgG repertoire. Of note, we detected high variability in the repertoire composition within each cohort. As a consequence, some individuals might still have better chances of developing certain bNAb classes because of other predispositions, e.g. by carrying specific gene segment alleles that encode for critical contact sites, as it has been previously demonstrated for VRC01-like bNAbs[53].

Our approach is restricted by the following limitations. First, there is an imbalance in the cohorts in terms of age and sex with chronically infected individuals being older and mostly (HIV-1) or exclusively (HCV) male. Second, we were using 2x300 bp sequencing and processed paired reads only if they overlapped by at least six nucleotides. Longer CDRH3 sequences are more likely to fail during this assembly[88], which could lead to an underestimation of $P_{gen}$ for long CDRH3 bNAbs. As a consequence, long CDRH3 antibodies such as the V2-apex or MPER class might be more likely generated than proposed by our current models. Third, we relied on the IMGT database for allelic variant calling and did not determine subject specific alleles de novo[89]. Novel allelic variants will therefore be counted as mutations and the total number of somatic mutations in donors with novel allelic variants might be slightly overestimated. In terms of $P_{SHM}$, this would translate to a higher probability of finding a particular mutation at a certain position. Finally, by performing bulk sequencing, we lose heavy-light chain pairing information, which could have an influence on the total probability of bNAbs and differ across cohorts. It will therefore be interesting to adapt the presented approach to paired sequencing techniques (e.g., 10X genomics) in the future.

Taken together, we present a framework for assessing the overall probability to develop an antibody with a specific CDR3 motif and pattern of point mutations. Our modeling approach on HIV-1-neutralizing antibodies quantitatively confirms that bNAbs are unlikely outcomes of the B cell evolution within a host due to individual or a combination of improbable sequence features. Using neutralization data and probability scores, we demonstrate that the more unlikely bNAbs tend to have a higher neutralization efficacy. Moreover, this approach allows us to identify potent antibodies with higher chances to be elicited by vaccination. Finally, the data suggest that chronic infection has no impact on the generation of bNAb sequences within the IgG[+] B cell memory compartment, fostering the hope that a potent vaccine should be able to elicit bNAbs in uninfected individuals.

## Methods
### Sample collection
Samples were obtained under study protocols approved by the Ethics Committees of the Medical Faculties of the University of Cologne and University of Bonn (study protocols 16-054 and 017/16, respectively; Clinical trial registration DRKS00010169). Recruitment and sample collection of uninfected control and HIV-1 cohorts was conducted in cologne, HCV cohort recruitment and sample collection was conducted in Bonn. All participants provided written informed consent

and received a financial compensation for their participation. Sex/gender was not considered in the design of the biosample collection protocol and samples were collected irrespective of sex and gender. Sex was assigned based on data recorded in the hospital system (male: 71; female: 31; n/a: 1).

## Serum IgG isolation
Serum samples from HIV-1-infected individuals were heat-inactivated at 56 °C for 40 min and incubated with Protein G Sepharose (GE Life Sciences) overnight at 4 °C. IgGs were eluted from chromatography columns using 0.1 M glycine (pH = 3.0) into 0.1 M Tris (pH = 8.0). Buffer was exchanged to PBS through Amicon 30 kDa spin membranes (Millipore). Concentrations of purified IgGs were determined by UV/Vis spectroscopy (A280) on a Nanodrop 2000 and samples were stored at 4 °C.

## Serum IgG neutralization test
Neutralization assays with serum IgGs against the 12-strain global virus panel, were performed in 96-well plates following published protocols[9,90]. To this end, 12 HIV-1 pseudovirus strains were each mixed with 1:2 serial dilutions of purified IgG (1 mg/ml starting concentration, 8 dilutions) and incubated for 1 h at 37 °C. TZM-bl cells (RRID:CVCL_B478; ordered from the HIV Reagent Program, Cat-No. ARP-8129) were added ($10^4$ per well) in growth medium (DMEM, Gibco; 10% heat-inactivated FBS, Sigma-Aldrich; 2 mM L-glutamine, Thermo Fisher; 1 mM sodium pyruvate, Gibco; 50 μg/ml gentamicin, Sigma-Aldrich; 25 mM HEPES, Biochrom) with DEAE-dextran at a final concentration of 10 μg/ml and incubated for 2 days. Equal amounts of Luciferin-containing lysis buffer (10 mM $MgCl_2$, 0.3 mM ATP, 0.5 mM Coenzyme A, 17 mM IGEPAL (all Sigma-Aldrich), and 1 mM D-Luciferin (GoldBio) in 200 mM Tris-HCL pH 7.8) was added and after 2 min incubation samples were resuspended and luminescence was measured with a luminometer (Berthold TriStar² LB942). For $IC_{50}$ determination, the background signal (non-infected TZM-bl cells) was subtracted and IgG concentrations resulting in a 50% RLU reduction compared to untreated virus control wells were determined by using murine leukemia virus (MuLV)-pseudotyped virus as a control for unspecific activity. All samples were tested in duplicates.

## Isolation of B cells and RNA Isolation
PBMCs were isolated by standard density gradient centrifugation using Histopaque (Sigma Aldrich) and LeucoSep tubes (Greiner Bio-one). Cells were stored at −150 °C in 90% (v/v) FBS (Sigma Aldrich) and 10% (v/v) DMSO (Sigma Aldrich). Plasma was collected and stored separately at −80 °C. B cells were enriched from PBMCs with CD19 microbeads (Miltenyi Biotec). B cells were stained with anti-human AF700-CD20 (clone 2H7, BD Biosciences Cat-No. 560631, RRID: AB_1727447; 1:80), APC-IgG (clone G18-145, BD Biosciences Cat-No. 550931, RRID: AB_398478; 1:20), PE-Cy-7-IgD (clone IA6-2, BD Biosciences Cat-No. 561314, RRID: AB_10642457; 1:20), FITC-IgM (clone G20-127, BD Biosciences Cat-No. 555782; RRID: AB_396117; 1:5), PerCP-Cy5.5-CD27 (clone M-T271, BD Biosciences Cat-No. 560612, RRID: AB_1727457; 1:5) or PE-CD27 (clone M-T271, BD Biosciences Cat-No. 560985, RRID: AB_395834; 1:40) and DAPI (Thermo Fischer, Cat-No. D1306; 3 μM). All antibodies are routinely tested by the vendor (BD Biosciences). $CD20^+IgG^+$ or $CD20^+IgD^+IgM^+CD27^-IgG^-$ B cells were sorted into FBS (Sigma-Aldrich) using a FACSAria Fusion cell sorter (BD Biosciences). Spike-in experiments (Supplementary Fig. S3) were performed with the human cell lines RAMOS (RRID: CVCL_0597; DSMZ, Cat-No. ACC 603), MEC-1 (RRID: CVCL_1870; DSMZ, Cat-No. ACC 497), SuDHL-5 (RRID: CVCL_1735; DSMZ, Cat-No. ACC 571), and RI-1 (RRID: CVCL_1885; DSMZ, Cat-No. ACC 585). RNA-Isolation was performed using the RNeasy Micro Kit (Qiagen) on a QiaCube (Qiagen) instrument.

## RT-PCR and next generation sequencing
BCR repertoire sequence data was generated by template-switch RT-PCR. cDNA was generated from 10 μl RNA according to the SMARTer RACE 5′/3′ manual using SMARTScribe Reverse Transcriptase (Takara) and a self-designed template-switch oligo (AGGGCAGTCAGTCG-CAGNNNNWSNNNNWSNNNNWSGCrGrGrG). cDNA was diluted with 10 μl Tricine-EDTA buffer (Takara) according to the manual. Heavy and light chain variable regions were pre-amplified from 5 μl cDNA each by PCR with 1 μM forward primer (CTGATACGATTCACGCTAGGG CAGTCAGTCGCAG) and 0.33 μM constant region-specific reverse primers (IgM: ATGGAGTCGGGGAAGGAAGTC, IgG: AGGTGTGCACGCC GCTGGTC, IgK: GGTGACTTCGCAGGCGTAG, IgL: GCCGCGTACTT GTTGTTGC) in a 30 μl reaction with Q5 DNA polymerase (New England Biolabs). Cycling conditions were one cycle 98 °C/30 s, four cycles 98 °C/10 s and 72 °C/30 s, four cycles 98 °C/10 s, 62 °C/30 s (IgG/IgM) or 68 °C (IgK/IgL)/30 s, and 72 °C/30 s, as well as a final extension cycle at 72 °C/5 min. PCR products were purified with a NucleoSpin Gel and PCR Clean-up Kit (Macherey Nagel, REF 740609) with a 1/6 dilution of NTI binding buffer in RNAse-free water. Samples were eluted in 15 μl elution buffer (Macherey Nagel). Heavy and light chain amplicons were enriched from 5 μl purified pre-amplification product by a nested PCR with 0.33 μM forward primer (IgG: NNNNCACGCTAGGGCAGTCAG; IgM: NNNNNCACGCTAGGGCAGTCAG; IgK/IgL: NNNNCACGC-TAGGGCAGTCAG) and 0.33 μM nested reverse primers (IgG: NNNNNSGATGGGCCCTTGGTGGARGC; IgM: NNNNGGTTGGGGCG GATGCACTCC; IgK: NNNNNNGGGAAGATGAAGACAGATGGT, IgL: NNNNNNGGGYGGGAACAGAGTGACC) with Q5 DNA Polymerase (New England Biolabs) in a 100 μl reactions. PCR conditions were one cycle 98 °C/30 s, five cycles 98 °C/10 s and 72 °C/30 s, five cycles 98 °C/10 s and 70 °C/30 s, 17 cycles 98 °C/10 s, 68 °C/30 s (IgG/IgM) or 62 °C (IgK/IgL)/30 s, and 72 °C/30 s, one final extension cycle at 72 °C/5 min. Amplicons were separated on a 1% agarose gel, purified with a NucleoSpin Gel and PCR Clean-up Kit (Macherey Nagel, REF 740609) and subjected to library preparation and Illumina MiSeq 2x300 bp sequencing (MiSeq Reagent Kit v3 600-cycle, Cat. MS-102-3003) at the Cologne Center for Genomics sequencing core facility.

## NGS data processing and sequence annotation
Data pre-processing was performed with a Python (v.3.6)-based pipeline, including python packages Biopython (v.1.78), pandas (v.0.23.4), NumPy (v.1.19.2), Matplotlib (v.3.3.4), and python-Levenshtein (v.0.12.2). Raw NGS reads were filtered for a mean Phred score of 25 and read-lengths of at least 250 bp. Reads were annotated with IgBLAST[91] to identify read-orientations and CDR3 sequences. Reads were then grouped by the UMI and wrongly assigned reads (collisions) were identified by comparison of V gene segment calls and a second 18-nucleotide molecular identifier within the CDR3. Next, reads were aligned with Clustal Omega[92,93] and consensus sequences were generated by taking into account the quality score-weighted base-calls for each position. The 2x300 bp consensus read pairs were then aligned and combined using the AssemblePairs.py script form the pRESTO toolkit with a minimal overlap of 6 nucleotides[94]. Annotation of pre-processed reads was performed by using BLAST (v.2.9) and IgBLAST(v.1.13)[91]. Reference templates for all functional (F) heavy and light chain V(D) J genes were retrieved from the IMGT database[95,96] (265 IGHV, 30 IGHD, 13 IGHJ, 66 IGKV, 9 IGKJ, 70 IGLV, and 7 IGLJ at the time point of this study). After sequence annotation with IgBLAST, sequences were filtered to include complete V/J annotations covering at least 250 nucleotides of the V gene as well as productive sequences (i.e., no STOP codon and in-frame CDR3 recombination, as determined by IgBLAST) only. To minimize the influence of sequencing and PCR errors, annotated sequences were only used for downstream analyses, when their UMIs were initially found in at least three reads (high-quality reads). For learning $P_{gen}$ and $P_{SHM}$ models, sequences

were additionally filtered out, when they contained gaps (i.e. insertions/deletions).

## Antibody selection

Antibody information was retrieved from the CATNAP database[59]. At the time of this study, the available antibody dataset comprised 507 entries, from which non-human antibodies, polyclonal antibodies, antibody mixtures, as well as mutants/chimeras and non-antibody proteins were removed to yield a final set of 291 antibodies. The 291 antibodies were tested against different or only partially overlapping viral panels (such as the 12-strain global panel[65] or the 118-strain multi-clade panel[61]), making it difficult to compare them in terms of breadth and potency[97]. Of the 291 antibodies, 50 have been tested against the global panel and 19 against the 118 multi-clade panel. As a trade-off between number of antibodies and accuracy of breadth and potency, we selected a subset of 56 strains from the 118 multi-clade panel, which resembles its clade distribution, its tiered categorization, and yields similar values for breadth and potency (Supplementary Fig. 4, Supplementary Data 3). 70 out of 291 antibodies have been tested against this 56 strain panel, are published with complete heavy chain nucleotide sequences (including 64 light chains) and neutralized at least 1 of the 56 strains. Heavy and light chains were annotated with the same IgBLAST/IMGT-based pipeline as the NGS data. Five light chains could not be annotated and were excluded.

## Determination of antibody breadth, mean potency and neutralization score

$IC_{50}$ values for individual antibody-virus combinations were derived from the CATNAP database[59] as the geometric mean of all published $IC_{50}$ values (i.e. from different studies). $IC_{50}$ values below the detection limit were set to an arbitrary threshold of 100 µg/ml. The mean potency of an antibody against a viral panel was determined by the geometric mean of all geometric mean $IC_{50}$ values that were below the arbitrary threshold of 100 µg/ml (also known as the mean potency of all neutralized strains). Antibody breadth was defined as the percentage of neutralized strains (i.e. geometric mean $IC_{50}$ above the arbitrary threshold of 100 µg/ml) to the total number of strains tested and is given as percentage. To determine the combined neutralization score, we took all $IC_{50}$ values that were below the threshold of 100 µg/ml for each antibody and sorted them from lowest to highest. We then divided the maximum coverage (100%) into 56 equally spaced increments (~1.786% coverage per strain), and assigned cumulative increments to the sorted $IC_{50}$ values, i.e. the lowest $IC_{50}$ is assigned to a coverage of 1.786%, the second lowest to 3.572%, and so on. Finally we plotted the $\log_{10}$ of the sorted $IC_{50}$ values against the cumulative coverage and determined the area under the curve. Low $IC_{50}$ values (i.e. highly potent antibodies) will increase the area by shifting the curves to the left, while higher breadth will increase the area by shifting the plateau of the curve to the top. For the ranking of bNAbs in Fig. 2b, the neutralization score is given as percentage of the highest score among the 70 antibodies. For the correlation of probabilities and neutralization (Fig. 2), the $\log_{10}$ of the neutralization score was used.

## IGoR inference and prediction of bNAb probabilities

Models for the probability of generation ($P_{gen}$) and somatic hypermutation ($P_{SHM}$) were learned by using the Inference and Generation Of Repertoires (IGoR) tool[63]. To reduce uncertainty in model inference and focus on comparison between cohorts, we pooled all productive sequences from all patients in a cohort. Productive sequences were chosen to explore the effect of possible selection effects on $P_{gen}$ and $P_{SHM}$ of a given BCR sequence. The somatic hypermutation (SHM) model is 5-mer based, taking into account the mutated position and its two neighbors on both sides. The model takes into account the full composition of these 5-mers and is context dependent. $P_{gen}$ generation models were consistent with models inferred on previously

published repertoires[58,98]. Indels in bNAbs were reverted to the most likely V and J templates according to the IgBLAST annotation before model building to improve alignment by IGoR. Masking of indels has no influence on $P_{gen}$ or $P_{SHM}$. To test correlations between the neutralization score and the overall likelihood of seeing a given bNAb, we defined a score $S$ for each bNAb as $S = c_1\log_{10}(P_{gen}) + c_2\log_{10}(P_{SHM})$. Coefficients $c_1$ and $c_2$ were determined by linear regression of score $S$ versus the $\log_{10}$ of the neutralization scores.

## Clones and trees

After V, D, and J annotation, unique sequences were partitioned into clones. First, the sequences were grouped into classes of identical V and J genes and equal CDR3 length. Within each class, clones were identified using single linkage clustering with a fixed threshold of 90% CDR3 nucleotide identity. Tree length was then estimated as the total number of unique mutations found in a given clone (a lower bound on the true tree length). For productive clones of more than 50 unique sequences tree topologies and branch lengths were inferred using RAxML with the GTRGAMMA model of nucleotide substitution[99]. Germline V and J genes were provided as an outgroup to aid the inference. To quantify the asymmetry of the phylogenies within each cohort we examined the distributions of two indices of imbalance, following the protocol of Nourmohammad et al. (2019)[100]. We compare the weight of the common ancestor of the clone $w_{anc}$ with the weights of its immediate descendants, $w_D$. The weight of a node is the number of leaves that stem from it and the ratio $w_D/w_{anc}$ quantifies the imbalance at the first branching. Additionally, we estimated the distribution of the ratio of mean terminal branch length to the mean length of all branches.

## Quantification and statistical analysis

Flow cytometry analysis and quantifications were done with FlowJo10 software. Statistical analyses were performed using GraphPad Prism (v8), Microsoft Excel for Mac (v14.7.3), Python (v3.6.8), and R (v4.0.0). For the identification of overlapping clonotypes in uninfected individuals a maximum of one amino acid length difference and three or less differences in absolute amino acid composition of CDR3s were considered as similar. To calculate the CDRH3 sequence overlap between replicates in the control experiment (Fig. 1 c), a random sample of $n = 3623$ (i.e. the size of the smallest dataset) was drawn from all unique CDRH3s of each set and the overlap (i.e. identical CDRH3 sequences) between these random samples was determined. Sampling and overlap determination was repeated 100 times and the mean overlap over all iterations was reported. Testing for significant differences between CDR3 length distributions and V gene mutation frequency distributions (Fig. 1d, e) was performed by global Kruskal-Wallis tests (stats.kruskal, Scipy v.1.5.2) and Dunn post-hoc tests (posthoc_dunn, scikit_posthocs v. 0.4.0 with 'holm' method for $p$ value adjustment). To estimate the fraction of spiked-in lymphoma B cells among naive B cells from an uninfected donor (Supplementary Fig. 3), the pairwise Levenshtein distance of all re-constituted CDRH3s was determined (python-Levenshtein v.0.12.2) in comparison to the most frequent cell line CDRH3. The observed distance frequencies revealed a bimodal distribution from which all comparisons with <4 amino acid distance (first peak) were counted as a B cell line CDRH3 and divided by all comparisons to get the fraction. For cluster analysis of the mixed cell sample (Supplementary Fig. 3), a network analysis was performed with the networkx python package (v.2.2). Each node represents a unique CDRH3. The node size is proportional to the frequency among all identified CDRH3s and nodes are connected, if they share at least 75% of their CDRH3 amino acid sequence. Nodes are colored according to the cell lines if they share at least 75% of the CDRH3 amino acid sequence with a cell line. Phylogenetic trees of viral panels (Supplementary Fig. 4) were generated and illustrated from aligned viral sequences from the CATNAP database[59] with Geneious Prime software

(v.2020.2.4, Jukes-Cantor genetic distance model and Neighbor-Joining method with 100 bootstrap replicates). Testing for significant differences in mean CDR3 lengths and mean V gene mutation frequencies (Fig. 3c, d, Supplementary Figs. 8 and 9) was performed by one-way ANOVA (stats.f_oneway, Scipy v.1.5.2) followed by a two-sided Tukey-HSD post hoc test (stats.multicomp.pairwise_tukeyhsd, statsmodels v.0.12.2). Differences in V gene segment usages (Fig. 3b, Supplementary Figs. 8 and 9) were investigated by individual Kruskal-Wallis tests (stats.kruskal, Scipy v.1.5.2) for each V gene segment with Bonferroni correction for multiple testing. In the case of significant differences in the global test, a Dunn post hoc test (posthoc_dunn, scikit_posthocs v.0.4.0 with 'holm' method for $p$ value adjustment) was performed for subgroup analysis.

## Data availability

The NGS data generated in this study have been deposited in the Sequence Read Archive (SRA) under the accession codes SAMN29624595 to SAMN29624713 [https://www.ncbi.nlm.nih.gov/sra?linkname=bioproject_sra_all&from_uid=857338] and the BioProject database under accession code PRJNA857338. The HIV-1 neutralizing antibody data used in this study are freely available in the Los Alamos HIV sequence database under their names as listed in Supplementary Data 4 [https://www.hiv.lanl.gov/components/sequence/HIV/neutralization/main.comp]. V/D/J reference data was received from the IMGT database [https://www.imgt.org/genedb/]. All remaining data that support the findings of this study are included in this article (and its supplementary information files). Requests for any additional data should be directed to the corresponding author and may be subject to restrictions based on data and privacy protection regulations and/or may require a Material Transfer Agreement (MTA). Requests will be responded to within 1–2 weeks. Source data are provided with this paper.

## Code availability

IGoR is freely available on the GitHub repository [https://github.com/statbiophys/IGoR]. A detailed description of the workflow, including the complete analysis pipeline with example data to run the code, is provided in the public GitHub repository [https://github.com/statbiophys/bnabs_prob], also available on Zenodo[101] [https://doi.org/10.5281/zenodo.8409733].

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

## Acknowledgements

We thank all members of the Klein Laboratory for support and helpful discussion, Viera Kovacova, Antonios Papadakis, Lukas Maas, and Milos Nikolic for advice on data evaluation and NGS pipeline programming, Peter Nürnberg, Janine Altmüller, and Christian Becker from the Cologne Center for Genomics (CCG) for sequencing support, Michael Lässig and Christa Stitz for support within the CRC1310, and Till Schoofs for assistance in antibody selection. This work was funded by grants from the German Research Foundation (DFG; CRC 1279 to F.K.; CRC 1310 to F.K., C.K., A.M.W., A.N., J.G., A.B.), the German Center for Infection Research (DZIF to P.S., F.K.), the European Research Council (ERC-StG639961 to F.K; CoG 724208 to A.M.W.), the Agence Nationale de la Recherche (ANR-19-CE45-0018 RESP-REP to T.M.), CAREER Award from the National Science Foundation (Grant No. 2045054 to A.N.), the MIRA award from the National Institutes of Health (Grant No. 1R35GM142795-01 to A.N.), and the DFG-Emmy Noether Program (Project No. 495793173 to P.S.).

## Author contributions

Conceptualization, C.K., T.M., A.M.W., and F.K.; methodology, C.K., C.L., M.S.E., N.S., J.G., A.B., L.D., A.N., T.M., A.M.W and F.K.; investigation, C.K., L.G., M.S.E.; resources, L.G., M.S., L.D., P.S., and H.G.; formal analysis, C.K., C.L., N.S., and J.G.; writing-original draft, C.K., T.M., A.M.W and F.K.; writing-reviewing and editing: all authors; visualization, C.K.; supervision, A.B., T.M., A.M.W., F.K.; funding acquisition, A.N., T.M., A.M.W., C.K., and F. K.

## Funding

## Competing interests

A patent application encompassing HIV-1 broadly neutralizing antibodies has been filed by the University of Cologne and lists P.S., H.G., and F.K. as inventors. P.S., H.G. and F.K. received payments from the University of Cologne for licensed HIV-1 broadly neutralizing antibodies. The remaining authors declare no competing interests.

## Additional information

[1]Laboratory of Experimental Immunology, Institute of Virology, Faculty of Medicine and University Hospital Cologne, University of Cologne, 50931 Cologne, Germany. [2]Laboratoire de physique de l'Ecole normale supérieure, CNRS, PSL University, Sorbonne Université, and Université Paris Cité, 75005 Paris, France. [3]German Center for Infection Research, Partner Site Bonn-Cologne, 50931 Cologne, Germany. [4]Excellence Cluster on Cellular Stress Responses in Aging Associated Diseases & Institute for Genetics, Faculty of Mathematics and Natural Sciences, University of Cologne, 50931 Cologne, Germany. [5]Department I of Internal Medicine, Faculty of Medicine and University Hospital Cologne, University of Cologne, 50937 Cologne, Germany. [6]Center for Molecular Medicine Cologne (CMMC), Faculty of Medicine and University Hospital of Cologne, University of Cologne, 50931 Cologne, Germany. [7]Department of Internal Medicine I, University Hospital of Bonn, Bonn, Germany. [8]German Center for Infection Research (DZIF), Partner Site Bonn-Cologne, Bonn, Germany. [9]Max Planck Institute for Dynamics and Self-Organization, Am Faßberg 17, 37077 Göttingen, Germany. [10]Department of Physics, University of Washington, 3910 15th Ave Northeast, Seattle, WA 98195, USA. [11]Department of Applied Mathematics, University of Washington, 4182 W Stevens Way NE, Seattle, WA 98105, USA. [12]Paul G. Allen School of Computer Science and Engineering, University of Washington, 85 E Stevens Way NE, Seattle, WA 98195, USA. [13]Fred Hutchinson Cancer Center, 1241 Eastlake Ave E, Seattle, WA 98102, USA. [14]Present address: Istituto Nazionale di Fisica Nucleare (INFN), Sezione di Roma I, 00185 Rome, Italy. [15]These authors contributed equally: Christoph Kreer, Cosimo Lupo, Meryem S. Ercanoglu. [16]These authors jointly supervised this work: Thierry Mora, Aleksandra M. Walczak, Florian Klein. ✉e-mail: florian.klein@uk-koeln.de

