## [Peer Review File · Nature Communications]

Probabilities of developing HIV-1 bNAb sequence features in uninfected and chronically infected individualsREVIEWER COMMENTS

Reviewer #1 (Remarks to the Author):

In their manuscript, Kreer et al. perform B cell receptor repertoire sequencing on cohorts of HIV and HCV infected and uninfected subjects, train computational models of VDJ recombination and somatic hypermutation, and then use these models to compute generation probabilities of antibodies both at the single antibody and repertoire levels. The authors first validated their NGS sequencing approach on a set of biological and technical replicates in three subjects, observing subject-specific patterns of CDRH3 lengths and mutation frequencies between individuals, but highly similar patterns within biological replicates. After repertoire sequencing their large cohorts, they curate a set of broadly neutralizing HIV antibodies and compare their sequence characteristics to the distributions of sequence characteristics from that of the uninfected controls to show that bnAbs tend to be more mutated and use longer CDRH3s. While this is confirmatory with what has been known about HIV bnAbs, it is nonetheless a well-presented and comprehensive analysis using an up-to-date bnAb set. They then use their software IgOR to learn models of recombination and somatic hypermutation from the uninfected NGS dataset and apply those to calculate generation probabilities for antibody sequences. A key finding of their work is that a linear combination of recombination and SHM probabilities correlates with bnAb neutralization capacity. The authors point out that outliers from that correlation that have high neutralization capacity and high probability (relatively speaking) make good targets for vaccine design, as they may be easier to elicit with a vaccine. The authors go on to show that at the repertoire level, very little difference is observed in sequence characteristics between HIV and HCV infected and uninfected subjects although mutation patterns in heavy chains skew to lower probabilities in the infected cohorts. Finally, they retrain the IgOR models on the infected subjects' repertoires and show no differences in the probabilities of bnAbs when trained on infected vs. uninfected datasets suggesting that elicitation of bnAbs is not predicated upon repertoire-level conditions that are specific to chronic infection. Overall, this is an important and comprehensive study that by combining rigorous and deep sequencing with computational models of antibody generation reveals insightful findings about the stochastic nature of bnAb development.

Major comments:

There are some limitations to the sequencing approach that the authors should state. 1) It has been shown that 5'RACE with 2x300 Illumina MiSeq sequencing can be limited in its ability to recover heavy chain reads with long CDRH3s especially for reads with longer 5'UTRs (Bernat et al. Front Imm 2019). The authors should discuss whether they think this introduces a bias in the CDRH3 length distributions that they report. 2) The authors use bulk heavy or light chain repertoire sequencing to reach the depth needed for a representative sample. However, the use of unpaired sequencing could miss important differences in the cohort's repertoires that are dependent upon the heavy and light chain pairing and the authors should acknowledge that limitation. 3) Novel subject-specific Ig gene allelic variants will be recorded as mutations in the authors' analysis, while perhaps this will only have a minor effect on the P_SHM calculation, this is another limitation that should be acknowledged.

The authors state: "Our analysis illustrates that the degree of improbable mutations is correlated with the binding site and is not a common feature of bnAbs per se". This statement is poorly worded and poorly supported by the authors' data. In Figure 2F, a great majority of bnAbs have a P_SHM of <-100 which is far to the left of the tail of the healthy distribution. Doesn't that data support a conclusion that improbable mutation patterns are indeed a common feature of bnAbs?

The MPER class of antibodies is well represented in the upper right panel of Figure 2G which is the quadrant of high neutralizers with higher probability. The authors should discuss whether the role of autoreactivity, which is associated with the MPER antibodies and which is not captured in the authors' generation probabilities, would explain over-representation

of MPER antibodies in this quadrant and would explain the relative dearth of MPER bnAbs elicited by HIV infection.

Minor comments:

The authors use V gene germline identity to report mutation loads but it's typical in the field to use the converse metric of mutation frequency (which can be defined for just the V gene if that is the specific region of interest) and it would be easier to put the results in the context of the previous wealth of literature on HIV bnAbs if the authors used mutation frequency instead.

The use of the term "healthy" should be avoided as it is a general and subjective term. The authors clearly mean HCV and HIV negative, so the term "uninfected" is a better choice.

Figure 1: The lower panel of 1C is not explained well in the legend and as far as I can tell, the lower panel of 1C is not referenced in the text.

Figure 2: The r value in 2G right panel is different from the r value in the text. The authors should address this discrepancy.

Figure 3: There is imbalance in the cohorts with regards to age (control vs. HIV-1/HCV) and sex (HCV/HIV vs control) and that should be mentioned as a study limitation.

Panels 3C and D are very small with thin lines and it is very difficult at that panel size for me to tell how similar or different the distributions truly are. Panel 3F legend should explain what wD/Wanc ratio is, or at least point the reader to the methods section for the description there. I do not see a lot of overlap with the red (HIV) and blue (control) lines in 3F, yet the text referencing this panel says there is "no obvious difference".

In Figure 3G, and throughout the text, the term "lineage" is used and I think it would be clearer to use the term "clone". A lineage is mainly interpreted as a line of descent from a common ancestor whereas a clone is the entire set of descendants from the original unmutated B cell progenitor.

In Figure 4C, why does the number of lambda chains differ when Pgen is used from models trained on HCV vs. HIV (n=29 vs. n=21, respectively?).

Indels are challenging to model and are discarded from this generation probability analysis, but even a simple enumeration of indels and a comparison between the cohorts' repertoires is of high interest to the HIV field given their frequent role in bnAb maturation. I think it would strengthen the manuscript to include such a panel of the frequency of insertions and deletions even as a supplemental figure, to determine whether indels occur more frequently in repertoires from chronic infection vs. controls.

Reviewer #2 (Remarks to the Author):

This manuscript offers a very interesting study of sequence-associated characteristics of broadly neutralizing antibodies against HIV-1. At the same time, it is predominantly descriptive -- essentially quantifying the various characteristics with respect to structural facets of the virus. As such, it represents a useful compendium of information. However, the new insights gained are relatively modest, with the most substantive being the conclusion (from Figure 4) that the distribution of characteristics is similar between the different subject cohorts -- evidence against the proposed notion that chronic infection is required for broadly neutralizing antibody development. Unfortunately, in its current form the study leaves a lot on the table concerning potential further insights, which could raise the impact of this work to more commensurate with the journal criteria. A major example is the calculations in Figure 2 aimed at elucidating any relationships between IGOR genetic event model parameters (Pgen, Pshm) and neutralization capability. A simple neutralization score is used as the metric to correlate these parameters against, and the corresponding

results are not compelling. But there could be much more nuanced quantification of the neutralization data for which a genetic event model (whether IGOR or a modification thereof) could generate more significant findings -- especially since the IGOR model has already been published previously, and thus does not provide a novel advance in itself.

Reviewer #3 (Remarks to the Author):

The manuscript titled "Probabilities of HIV-1 bNAb development in healthy and chronically Infected individuals" by Christoph Kreer and colleagues performed unbiased NGS and established probabilistic models for SHM and recombination on the BCR repertoire from health individuals. By analyzing 75 broadly HIV-1-neutralizing antibodies utilizing the models established, the authors revealed that a combined score based on probabilities of generating a given CDRH3 and accumulating a unique pattern of point mutations is highly predictive of the neutralization capacity, and the less likely bNABs may be the most potent ones. They further compared the data from chronically infected patients and concluded that the global BCR repertoire features in chronic infections are very similar to healthy repertoires, and most importantly, they found that equal probabilities for bNAb development in healthy and chronically infected individuals. The results presented in this manuscript have important implications in vaccine development by targeted elicitation of antibodies of higher neutralization efficacy and higher elicitation probability. The paper is well written and the illustrations are clear and nicely done. I only have minor questions

- 1. When analyzing the Probability score S of control vs Probability score S of antibodies in Figure 4E/F and G, the CD4BS antibody CH103 and V3-glycan antibody 10-were outliers to some extent, is there an explanation?**
- 2. Correlation has been identified between log(neutralization score) and the combined probability score, the authored showed that VRC01, N6, 561-01-18 and BG18 (Figure 2G, right panel), it seems that MEPR and V2 apex antibodies showed a reasonable combination of high potency and high probability while V3-glycan antibodies were less optimal target for vaccine design, and the CD4BS antibodies are high in potency but less probable, is this right?**

Point-by-point response

Kreer et al., "Probabilities of HIV-1 bNAb development in uninfected and chronically infected individuals", NCOMMS-23-03657A

Reviewer #1 (Remarks to the Author):

"In their manuscript, Kreer et al. perform B cell receptor repertoire sequencing on cohorts of HIV and HCV infected and uninfected subjects, train computational models of VDJ recombination and somatic hypermutation, and then use these models to compute generation probabilities of antibodies both at the single antibody and repertoire levels. The authors first validated their NGS sequencing approach on a set of biological and technical replicates in three subjects, observing subject-specific patterns of CDRH3 lengths and mutation frequencies between individuals, but highly similar patterns within biological replicates. After repertoire sequencing their large cohorts, they curate a set of broadly neutralizing HIV antibodies and compare their sequence characteristics to the distributions of sequence characteristics from that of the uninfected controls to show that bnAbs tend to be more mutated and use longer CDRH3s. While this is confirmatory with what has been known about HIV bnAbs, it is nonetheless a well-presented and comprehensive analysis using an up-to-date bnAb set. They then use their software IgOR to learn models of recombination and somatic hypermutation from the uninfected NGS dataset and apply those to calculate generation probabilities for antibody sequences. A key finding of their work is that a linear combination of recombination and SHM probabilities correlates with bnAb neutralization capacity. The authors point out that outliers from that correlation that have high neutralization capacity and high probability (relatively speaking) make good targets for vaccine design, as they may be easier to elicit with a vaccine. The authors go on to show that at the repertoire level, very little difference is observed in sequence characteristics between HIV and HCV infected and uninfected subjects although mutation patterns in heavy chains skew to lower probabilities in the infected cohorts. Finally, they retrain the IgOR models on the infected subjects' repertoires and show no differences in the probabilities of bnAbs when trained on infected vs. uninfected datasets suggesting that elicitation of bnAbs is not predicated upon repertoire-level conditions that are specific to chronic infection. Overall, this is an important and comprehensive study that by combining rigorous and deep sequencing with computational models of antibody generation reveals insightful findings about the stochastic nature of bnAb development."

We kindly thank Reviewer #1 for appreciating our work as "an important and comprehensive study" and for providing valuable comments that we thoroughly incorporated to further improve the manuscript. The specific points raised by this reviewer are addressed in detail below.

P1: "There are some limitations to the sequencing approach that the authors should state."

Response P1: We agree that the presented sequencing approach is subject to some limitations and therefore added a new paragraph to the discussion (**revised manuscript page 17, lines 405-423**) to address the limitations raised by this reviewer (see below).

P2: “1) It has been shown that 5’RACE with 2x300 Illumina MiSeq sequencing can be limited in its ability to recover heavy chain reads with long CDRH3s especially for reads with longer 5’UTRs (Bernat et al. *Front Imm* 2019). The authors should discuss whether they think this introduces a bias in the CDRH3 length distributions that they report.”

Response P2: We fully agree that this is a critical point for our conclusions and therefore added a discussion on a potential bias in CDRH3 length distributions (**revised manuscript page 17, lines 407-415**). In addition, we performed further analyses, which we provide to this reviewer. As published by Bernat et al., we found an enrichment of longer CDRH3s in the unassembled paired reads (**P2P-Fig. 1 a**), leading to their exclusion from the analysis (**P2P-Fig. 1 a and b**). The impact of excluded sequences on the overall distributions was marginal (15.0 vs. 15.0, 15.0 vs. 15.0, 16.0 vs. 16.0, and 15.5 vs. 16.0, for the four cohort median lengths, respectively; **P2P-Fig. 1 c**) and the fraction of excluded long CDRH3s (≥ 20 amino acids) was only slightly and not significantly different in untreated HIV-1- or HCV-infected individuals ($p = 0.291$ for a global One-way ANOVA; **P2P-Fig. 1 b**). Moreover, despite the partial exclusion of some CDRH3s, we still detected a shift towards longer mean CDRH3 lengths in HCV- and untreated HIV-1-infected individuals (**Fig. 3 c, Supplementary Fig. 8, P2P-Fig. 1 d**). Therefore, P_{gen} could be slightly underestimated for long CDRH3 bNAbs, but comparably for all cohorts and does not change any of our conclusions.

P2P-Figure 1: Impact of excluded unassembled sequences on CDRH3 length distributions. (a)

Reads that failed the assembly pipeline were annotated and filtered by the same criteria as the originally included data in the manuscript. CDRH3 length distributions were calculated for each individual and mean distributions were plotted together with the original distributions from the manuscript (colored by cohort). (b) The fraction of excluded CDRH3s was calculated for each individual (dots) with respect to all (left panel) or long CDRH3s (≥ 20 amino acids, right panel) and grouped by cohorts (colors as in a). Statistical testing was performed by One-way ANOVA ($F[3,99] = 1.264$, $p = 0.291$). (c) Excluded CDRH3s were combined with the original data (black dashed distributions) and plotted together with the original distributions (colored by cohort). Vertical dashed lines and numbers in (a) and (b) depict the median CDRH3 length. (d) Mean CDRH3 lengths were determined for each individual for the originally included (left panel) or the combination of included and excluded data (right panel) and plotted for each cohort (colored dots). Boxplots in (b) and (d) depict 25-75% quantiles with medians as lines and min/max values as whiskers. aa: amino acids.

P3: “2) The authors use bulk heavy or light chain repertoire sequencing to reach the depth needed for a representative sample. However, the use of unpaired sequencing could miss important differences in the cohort’s repertoires that are dependent upon the heavy and light chain pairing and the authors should acknowledge that limitation.”

Response P3: We thank Reviewer #1 for his/her comment and now acknowledge the lack of chain pairing as a limitation in the discussion (**revised manuscript page 17, lines 420-423**).

P4: “3) Novel subject-specific Ig gene allelic variants will be recorded as mutations in the authors’ analysis, while perhaps this will only have a minor effect on the P_SHM calculation, this is another limitation that should be acknowledged.”

Response P4: We fully agree with Reviewer #1 and added a discussion of the impact of allelic variants on P_SHM determination to the limitation paragraph (**revised manuscript page 17, lines 415-420**).

P5: “The authors state: “Our analysis illustrates that the degree of improbable mutations is correlated with the binding site and is not a common feature of bnAbs per se”. This statement is poorly worded and poorly supported by the authors’ data. In Figure 2F, a great majority of bnAbs have a P_SHM of <-100 which is far to the left of the tail of the healthy distribution. Doesn’t that data support a conclusion that improbable mutation patterns are indeed a common feature of bnAbs?”

Response P5: We thank Reviewer #1 for pointing this out and rephrased the sentence to highlight that different classes of bNAbs strongly vary in their P_{SHM} values (**revised manuscript page 14, lines 349-351 and page 15, lines 355-356**).

P6: “The MPER class of antibodies is well represented in the upper right panel of Figure 2G which is the quadrant of high neutralizers with higher probability. The authors should discuss whether the role of autoreactivity, which is associated with the MPER antibodies and which is not captured in the authors’ generation probabilities, would explain over-representation of MPER antibodies in this quadrant and would explain the relative dearth of MPER bnAbs elicited by HIV infection.”

Response P6: We thank Reviewer #1 for this valuable comment. The predicted higher probabilities of MPER antibodies despite their auto-reactive nature is indeed a very interesting point, which we now discuss in detail in the revised manuscript (**revised manuscript pages 15-16, lines 372 -389**).

Minor comments:

P7: “The authors use V gene germline identity to report mutation loads but it’s typical in the field to use the converse metric of mutation frequency (which can be defined for just the V gene if that is the specific region of interest) and it would be easier to

put the results in the context of the previous wealth of literature on HIV bnAbs if the authors used mutation frequency instead.”

Response P7: We agree with the reviewer and now report all V gene germline identities as nucleotide mutation frequencies in % throughout the manuscript, figures and tables (**Figures 1, 2, 3, S5, S8, S9, and Supplementary Table 4**).

P8: “The use of the term “healthy” should be avoided as it is a general and subjective term. The authors clearly mean HCV and HIV negative, so the term “uninfected” is a better choice.”

Response P8: To be more precise, the term “healthy” has been replaced by “uninfected” throughout the manuscript and figures, as suggested by this reviewer.

P9: “Figure 1: The lower panel of 1C is not explained well in the legend...”

Response P9: We thank the reviewer for carefully reading the figure legends and rephrased the explanation of the lower panel of **Fig. 1c** in the figure legend as well as included a reference to the methods part, where we provide the detailed description of how the plot was generated (**Fig. 1 legend**).

P10: “...and as far as I can tell, the lower panel of 1C is not referenced in the text.”

Response P10: We now explicitly refer to “Fig. 1c, lower panel” (**revised manuscript page 6, line 139**) to make it easier to correlate the text passages with the respective figure panels.

P11: “Figure 2: The r value in 2G right panel is different from the r value in the text. The authors should address this discrepancy.”

Response P11: We understand the confusion and now clarified in the text that we refer to either the spearman correlation coefficients (in the plots) or the r and p values from the linear regression of the probability score S (**Fig. 2 legend; revised manuscript page 10, lines 231-233 and 235-237**).

P12: “Figure 3: There is imbalance in the cohorts with regards to age (control vs. HIV-1/HCV) and sex (HCV/HIV vs control) and that should be mentioned as a study limitation.”

Response P12: We thank Reviewer #1 for pointing this out and now report imbalances in the cohorts as a limitation in the discussion (**revised manuscript page 17, lines 405-407**).

P13: “Panels 3C and D are very small with thin lines and it is very difficult at that panel size for me to tell how similar or different the distributions truly are.”

Response P13: For better visualization, we re-organized Fig.3 panels c and d, including the rescaling of the distribution plots and an increased line thickness (**Fig. 3c, d**). However,

we also want to highlight that light chain CDR3 length distributions are almost identical across individuals and cohorts. As a consequence, standard deviations of the distributions (shaded areas) are merely visible for light chains, even after rescaling the plots.

P14: *“Panel 3F legend should explain what w_D/w_{anc} ratio is, or at least point the reader to the methods section for the description there.”*

Response P14: We thank this reviewer for recognizing this missing information and added an explanation of the w_D/w_{anc} ratio to the figure legend (**Fig. 3f**).

P15: *“I do not see a lot of overlap with the red (HIV) and blue (control) lines in 3F, yet the text referencing this panel says there is “no obvious difference”.”*

Response P15: We understand that this might be misleading. The distribution is plotted on a log scale. Therefore, most of the w_D/w_{anc} ratios are either <0.25 or ≥ 0.75 for all three cohorts. The differences in the middle of the plot are actually small, since this represents a region of the log scale where numbers are low, i.e., less reliable, as can be seen from the jaggedness. It is important that all distributions are similar in shape/symmetry and have the majority of data points in the same regions (<0.25 and ≥ 0.75), which we now clarified in the text (**revised manuscript page 11, lines 269-270**).

P16: *“In Figure 3G, and throughout the text, the term “lineage” is used and I think it would be clearer to use the term “clone”. A lineage is mainly interpreted as a line of descent from a common ancestor whereas a clone is the entire set of descendants from the original unmutated B cell progenitor. “*

Response P16: We agree with this reviewer on the appropriate terminology and replaced "lineage" by "clone" throughout the manuscript.

P17: *“In Figure 4C, why does the number of lambda chains differ when Pgen is used from models trained on HCV vs. HIV (n=29 vs. n=21, respectively?).“*

Response P17: We thank this reviewer for carefully checking the figures and identifying the discrepancy in light chains, which we traced back to partially missing sequencing data. We therefore re-analyzed all cohorts for the revised manuscript and updated the corresponding figures (**Figs. 2 and 4; Supplementary Figs. 5 and 6**), tables (**Supplementary Table 5**) and text passages (**revised manuscript page 13, lines 311-316, as well as pages 13-14, line 327-328**). Importantly, none of the changes has any impact on our conclusions.

P18: *“Indels are challenging to model and are discarded from this generation probability analysis, but even a simple enumeration of indels and a comparison between the cohorts’ repertoires is of high interest to the HIV field given their frequent role in bnAb maturation. I think it would strengthen the manuscript to include such a panel of the frequency of insertions and deletions even as a supplemental figure, to*

determine whether indels occur more frequently in repertoires from chronic infection vs. controls.”

Response P18: We thank the reviewer for this valuable suggestion and added the frequencies of insertions and deletions for the three cohorts as a novel supplementary figure and to the text (**Supplementary Fig. 7, revised manuscript page 11, lines 264-267**).

Reviewer #2 (Remarks to the Author):

“This manuscript offers a very interesting study of sequence-associated characteristics of broadly neutralizing antibodies against HIV-1. At the same time, it is predominantly descriptive -- essentially quantifying the various characteristics with respect to structural facets of the virus. As such, it represents a useful compendium of information. However, the new insights gained are relatively modest, with the most substantive being the conclusion (from Figure 4) that the distribution of characteristics is similar between the different subject cohorts -- evidence against the proposed notion that chronic infection is required for broadly neutralizing antibody development. Unfortunately, in its current form the study leaves a lot on the table concerning potential further insights, which could raise the impact of this work to more commensurate with the journal criteria. “

We thank this reviewer for critically evaluating our manuscript and for appreciating that our manuscript *“offers a very interesting study of sequence-associated characteristics of broadly neutralizing antibodies against HIV-1”*. This reviewer also raises the concern that the study *“is predominantly descriptive”* and *“new insights gained are relatively modest”*. We therefore want to highlight again that beyond dataset description, we learned computational models and present both a comprehensive neutralization score and a combined probability score that allow us for the first time to (i) formally proof that bNAbs are on average better neutralizer the less likely they are to develop, (ii) identify bNAbs that are more likely to be generated, which has major implications for vaccine approaches, and (iii) show that chronic infection is not a prerequisite for the development of HIV-1 targeting bNAbs, which has not been demonstrated before.

“A major example is the calculations in Figure 2 aimed at elucidating any relationships between IGOR genetic event model parameters (P_{gen} , P_{shm}) and neutralization capability. A simple neutralization score is used as the metric to correlate these parameters against, and the corresponding results are not compelling. But there could be much more nuanced quantification of the neutralization data for which a genetic event model (whether IGOR or a modification thereof) could generate more significant findings -- especially since the IGOR model has already been published previously, and thus does not provide a novel advance in itself.”

We thank the reviewer for this comment. While other factors such as Fc-mediated effector functions, including antibody-dependent cellular cytotoxicity (ADCC), antibody-dependent cellular phagocytosis (ADCP), and complement-dependent cytotoxicity (CDC), certainly

contribute to the HIV-1 antibody response, we focused on the two most important (and most frequently reported) characteristics of antibody-mediated virus neutralization, i.e., breadth and potency. Since these values strongly depend on the viral panel they have been assessed with, we thoroughly selected a 56-strain subpanel to guarantee comparability between the bNAbs in this study. Moreover, we aimed to directly rank and compare bNAbs with regard to the overall neutralization capacity. To better convey the challenges associated with comparing bNAbs and the advantages of introducing a combined neutralization score, we elaborated on these aspects in the manuscript (**revised manuscript page 8, lines 177-189**).

In addition, as the reviewer pointed out, IGoR has already been published previously. We certainly did not intend to claim that IGoR is a novel advance of this study, which we therefore clarified again in the results section (**revised manuscript page 9, line 205**). However, we want to highlight that we applied this software (in combination with other tools such as IgBLAST) as one part of our analysis pipeline on a large dataset of newly generated NGS-repertoire data from 103 uninfected or chronically-infected individuals to (i) learn novel cohort specific models, (ii) determine cohort specific probabilities of a comprehensive panel of bNAbs, and (iii) draw biological conclusions from them (see previous response). Importantly, for the first time, we combine parameters from IGoR in a global probability score that takes into account both, VDJ recombination and SHM. Beyond the presented biological findings, we also want to encourage the community to apply and adapt our framework and therefore added a github repository in the "CODE AVAILABILITY" section, where we provide the full analysis pipeline, together with test data and detailed installation and execution explanations (**revised manuscript pages 27-28, lines 667-671**).

Reviewer #3 (Remarks to the Author):

“The manuscript titled “Probabilities of HIV-1 bNAb development in healthy and chronically Infected individuals” by Christoph Kreer and colleagues performed unbiased NGS and established probabilistic models for SHM and recombination on the BCR repertoire from health individuals. By analyzing 75 broadly HIV-1-neutralizing antibodies utilizing the models established, the authors revealed that a combined score based on probabilities of generating a given CDRH3 and accumulating a unique pattern of point mutations is highly predictive of the neutralization capacity, and the less likely bNAbs may be the most potent ones. They further compared the data from chronically infected patients and concluded that the global BCR repertoire features in chronic infections are very similar to healthy repertoires, and most importantly, they found that equal probabilities for bNAb development in healthy and chronically infected individuals. The results presented in this manuscript have important implications in vaccine development by targeted elicitation of antibodies of higher neutralization efficacy and higher elicitation probability. The paper is well written and the illustrations are clear and nicely done. I only have minor questions.”

We kindly thank this reviewer for appreciating the presentation of our study as well as the implications of the presented findings. This reviewer had two questions, which we have addressed in detail below.

P1: “1. When analyzing the Probability score S of control vs Probability score S of antibodies in Figure 4E/F and G, the CD4BS antibody CH103 and V3-glycan antibody 10-were outliers to some extent, is there an explanation?”

Response P1: We thank Reviewer #3 for carefully reviewing Figure 4. This reviewer is absolutely correct that these outliers are surprising, which was due to the use of an incomplete sequence dataset for the probability calculations. For the revised manuscript, we therefore re-analyzed the probabilities with the complete set of available sequences and modified the manuscript accordingly (**Figs. 2 and 4; Supplementary Figs. 5 and 6; Supplementary Table 5**). As a result, the statistical power increased and we do no longer see any outliers, thus further supporting our initial conclusions (**revised manuscript page 13, lines 311-316, as well as pages 13-14, line 327-328**). Again, we would like to thank Reviewer #3 for making this important observation.

P2: “2. Correlation has been identified between log(neutralization score) and the combined probability score, the authored showed that VRC01, N6, 561-01-18 and BG18 (Figure 2G, right panel), it seems that MEPR and V2 apex antibodies showed a reasonable combination of high potency and high probability while V3-glycan antibodies were less optimal target for vaccine design, and the CD4BS antibodies are high in potency but less probable, is this right?”

Response P2: We thank this reviewer for highlighting these observations. The conclusions on the different epitope classes are correct and a very import point of this study. We therefore further emphasized them in the revised discussion (**revised manuscript pages 15-16, lines 370-389**)

REVIEWERS' COMMENTS

Reviewer #1 (Remarks to the Author):

The authors have comprehensively addressed all of my comments and the manuscript is notably strengthened by their revisions. The work is a tour de force combination of sequencing and probabilistic immunogenetic analysis providing keen insights into the B cell response to HIV infection with important implications for HIV vaccine design.

Reviewer #3 (Remarks to the Author):

The authors have greatly improved this manuscript, I have no more comment.

Point-by-point response

Kreer et al., "Probabilities of developing HIV-1 bNAb sequence features in uninfected and chronically infected individuals", NCOMMS-23-03657A

Reviewer #1 (Remarks to the Author):

"The authors have comprehensively addressed all of my comments and the manuscript is notably strengthened by their revisions. The work is a tour de force combination of sequencing and probabilistic immunogenetic analysis providing keen insights into the B cell response to HIV infection with important implications for HIV vaccine design."

We kindly thank Reviewer #1 for dedicating his/her precious time and providing highly valuable comments during the revision process that clearly strengthened this manuscript.

Reviewer #3 (Remarks to the Author):

"The authors have greatly improved this manuscript, I have no more comment."

We kindly thank Reviewer #3 for reviewing our manuscript and appreciating the improvements made.